# Inhibition of a K9/K36 demethylase by an H3.3 point mutation found in paediatric glioblastoma

Hsiao P.J. Voon[1], Maheshi Udugama[1], Wendi Lin[1], Linda Hii[1], Ruby H.P. Law[1], David L. Steer[2], Partha P. Das[3], Jeffrey R. Mann[4] & Lee H. Wong[1]

An array of oncogenic histone point mutations have been identified across a number of different cancer studies. It has been suggested that some of these mutant histones can exert their effects by inhibiting epigenetic writers. Here, we report that the H3.3 G34R (glycine to arginine) substitution mutation, found in paediatric gliomas, causes widespread changes in H3K9me3 and H3K36me3 by interfering with the KDM4 family of K9/K36 demethylases. Expression of a targeted single-copy of H3.3 G34R at endogenous levels induced chromatin alterations that were comparable to a KDM4 A/B/C triple-knockout. We find that H3.3 G34R preferentially binds KDM4 while simultaneously inhibiting its enzymatic activity, demonstrating that histone mutations can act through inhibition of epigenetic erasers. These results suggest that histone point mutations can exert their effects through interactions with a range of epigenetic readers, writers and erasers.

[1] Department of Biochemistry and Molecular Biology, Cancer Program, Biomedicine Discovery Institute, Monash University, Wellington Road, Clayton, Victoria 3800, Australia. [2] Biomedical Proteomics Facility, Monash University, Wellington Road, Clayton, Victoria 3800, Australia. [3] Anatomy and Developmental Biology, Monash University, Wellington Road, Clayton, Victoria 3800, Australia. [4] Genome Modification Platform, Monash University, Wellington Road, Clayton, Victoria 3800, Australia. Correspondence and requests for materials should be addressed to L.H.W. (email: lee.wong@monash.edu)

Eukaryotic chromosomes are comprised of a repeated array of approximately 147 bp of DNA wrapped around a nucleosome. Each nucleosome consists of an octamer of four core histone proteins; two subunits each of H2A, H2B, H3, H4, or an equivalent variant. The DNA/nucleosome complex forms the basic unit of chromatin, a structure which confers genomic stability and protects against DNA damage. In addition to conferring structural integrity through direct interactions with DNA, nucleosomes can also be post-translationally modified (PTM) to facilitate dynamic and nuanced regulation of DNA-related processes. These PTMs include the trimethylation of key lysine residues on the N-terminal tail of histone H3 such as H3K9me3, H3K27me3 and H3K36me3, which are involved in promoter silencing and prevent spurious intragenic transcription[1,2]. The steady state equilibrium of these modifications is dependent on the balanced activity of chromatin modifiers such as methyltransferases (e.g. SETDB1, PRC2 and SETD2) [3–5] and demethylases (e.g. KDM4 A/B/C)[6,7]. The continual action of chromatin modifiers is required to maintain appropriate chromatin states to facilitate all genomic processes. Any significant disruption of chromatin pathways leads to destabilisation of the genome by interfering with DNA replication, repair and transcription, and ultimately renders genomic DNA vulnerable to mutagenic events.

This loss of genomic integrity is a hallmark of cancers and accordingly, mutations in chromatin-related genes have been identified as oncogenic drivers. Many cancers harbour at least one mutation in a chromatin-related gene including DNA methyltransferases, histone modifiers, chromatin readers and remodellers[8]. In addition, some oncogenic mutations can alter chromatin states indirectly through the inhibition of chromatin modifiers. For example, point mutations in a metabolic enzyme, isocitrate dehydrogenase (IDH1/2) produces 2-hydroxyglutarate (2-HG)[9–11], an oncometabolite that competitively inhibits 2-oxoglutarate (2-OG)-dependent dioxygenase histone and DNA demethylases[12,13]. The IDH1/2 mutations are frequent events in certain cancers such as lower grade gliomas[14–17] where they also overlap with inactivating mutations in ATRX[14–17], a chromatin remodeller which deposits histone H3.3 at heterochromatin[18–20]. H3.3 is a replication-independent histone variant, which replaces canonical histone H3.1/2 outside of S-phase[21]. Interestingly, point mutations in histone H3.3 are common events in certain cancers, indicating that the ATRX/H3.3 chromatin pathway is a key contributor to the molecular pathogenesis of particular cancers.

The oncohistone mutations which have been discovered to date, are uniformly single-copy substitution mutations predominantly on H3.3, with rare occurrences on H3.1/2. Examples include the H3/H3.3 K27M mutation which is frequently detected in paediatric gliomas[22,23], and H3/H3.3 K36M mutations which are common in chondroblastomas[24] and head and neck squamous cell carcinomas[25]. These lysine to methionine mutations have been demonstrated to act in a dominant-negative manner by inhibiting the histone lysine methyltransferases which normally methylate the cognate residues, resulting in genome-wide chromatin alterations[26,27]. In addition, a number of substitutions have been detected at the glycine residue at H3/H3.3 position 34 (G34) [22–24], including an H3.3 G34R substitution which is commonly found in paediatric gliomas[22,23]. The functional consequences associated with G34 mutations are not well understood. H3.3 G34R has been reported to directly inhibit SETD2-mediated (SET domain containing 2) H3.3 K36me3 on the mutated histone[26], though not in a dominant-negative manner which is expected from heterozygous single-copy mutations. Furthermore, oncogenic chromatin mutations generally trigger genome-wide chromatin alterations but the genomic changes associated with H3.3 G34R have not yet been established.

We created a single-copy H3f3a G34R targeted mutation in mouse embryonic stem (ES) cells to faithfully recapitulate the mutations identified in paediatric gliomas. We find that this mutation triggers a gain in H3K36me3 across the genome, suggesting that H3.3 G34R inhibits the activity of a histone lysine demethylase. We identify the KDM4 A/B/C family of K9/K36 demethylases as likely candidates, and show that the H3K9me3 and H3K36me3 profile in H3.3 G34R mutants closely resemble a KDM4 A/B/C triple-knockout. Furthermore, we demonstrate that H3.3 G34R preferentially binds to KDM4 and inhibits its enzymatic activity. Studies of other oncohistone mutations have described a general mechanism where the mutated residues inhibit the activity of a chromatin remodeller (e.g. K27M/PRC2, K36M/SETD2). We describe a similar phenomenon for the H3.3 G34R mutations and identify the KDM4 histone lysine demethylases as the key chromatin modifiers, which are disrupted by this mutant histone. The IDH1/2 mutations are also known to inhibit the KDM4 demethylases, raising the possibility that H3.3 G34R and IDH1/2 mutations may drive oncogenesis through parallel pathways.

## Results

**Creation of mouse ES cells with an H3.3 G34R mutation.** In mouse and humans, the H3.3 histone variant is expressed from two distinct genes (H3f3a and H3f3b) which encode identical proteins[28,29]. The G34R substitution in paediatric gliomas always occur as heterozygous mutations on H3f3a[22,23], and tumours retain three normal copies of H3.3 in addition to the canonical H3.1/2 histones. Therefore, the oncogenic properties of H3.3 G34R emerge in a context where the mutant histone comprises a small fraction of total cellular H3. As such, modelling the H3.3 G34R mutation with an ectopic overexpression system could result in artefactual phenotypes that mask the underlying oncogenic mechanisms. Instead, we opted to create a targeted mutation on the mouse H3f3a locus to accurately reflect the full H3.3 genotype of paediatric gliomas.

A two-step recombination strategy was used to express H3.3 G34R mutant transcript from the endogenous H3f3a promoter[28,29]. A targeting vector comprised of three cassettes (wild-type (WT) H3f3a, neo and H3f3a G34R) was substituted for exon 2 of H3f3a, creating a conditional allele which expressed a WT H3f3a minigene from the endogenous H3f3a promoter (Fig. 1a). Cre-recombinase was added to excise the WT H3f3a and neo cassettes from the conditional allele, and brought the H3f3a G34R transgene under the control of the endogenous H3f3a promoter (Fig. 1a). The correct targeting and Cre-mediated excision were confirmed by Southern blot using digests and probes as illustrated (Fig. 1a). Results from WT, conditional and two G34R mutants are shown (Fig. 1b); fragment sizes were consistent with predicted sizes for single-copy targeted integration (Fig. 1b). A schematic depiction of the full H3.3 genotype of H3f3a G34R, hereafter referred to as H3.3 G34R, is shown (Fig. 1c).

To ensure that the H3.3 G34R mutation was appropriately expressed, we used RNA-sequencing reads to assess relative expression of H3f3a G34R, WT H3f3a and H3f3b (Fig. 1c). As H3f3a and H3f3b are genetically distinct, RNA-seq reads which were informative for codon 34 were also able to discriminate between these two alleles (Supplementary Fig. 1). The relative proportion of reads which mapped to each H3.3 allele showed that H3f3a G34R is expressed at an appropriate level against a background of WT H3f3a and H3f3b (Fig. 1d). The full complement of informative RNA-seq reads are also shown with the distinguishing nucleotides highlighted (Supplementary Fig. 1). The expression of H3.3 G34R was further confirmed by Sanger

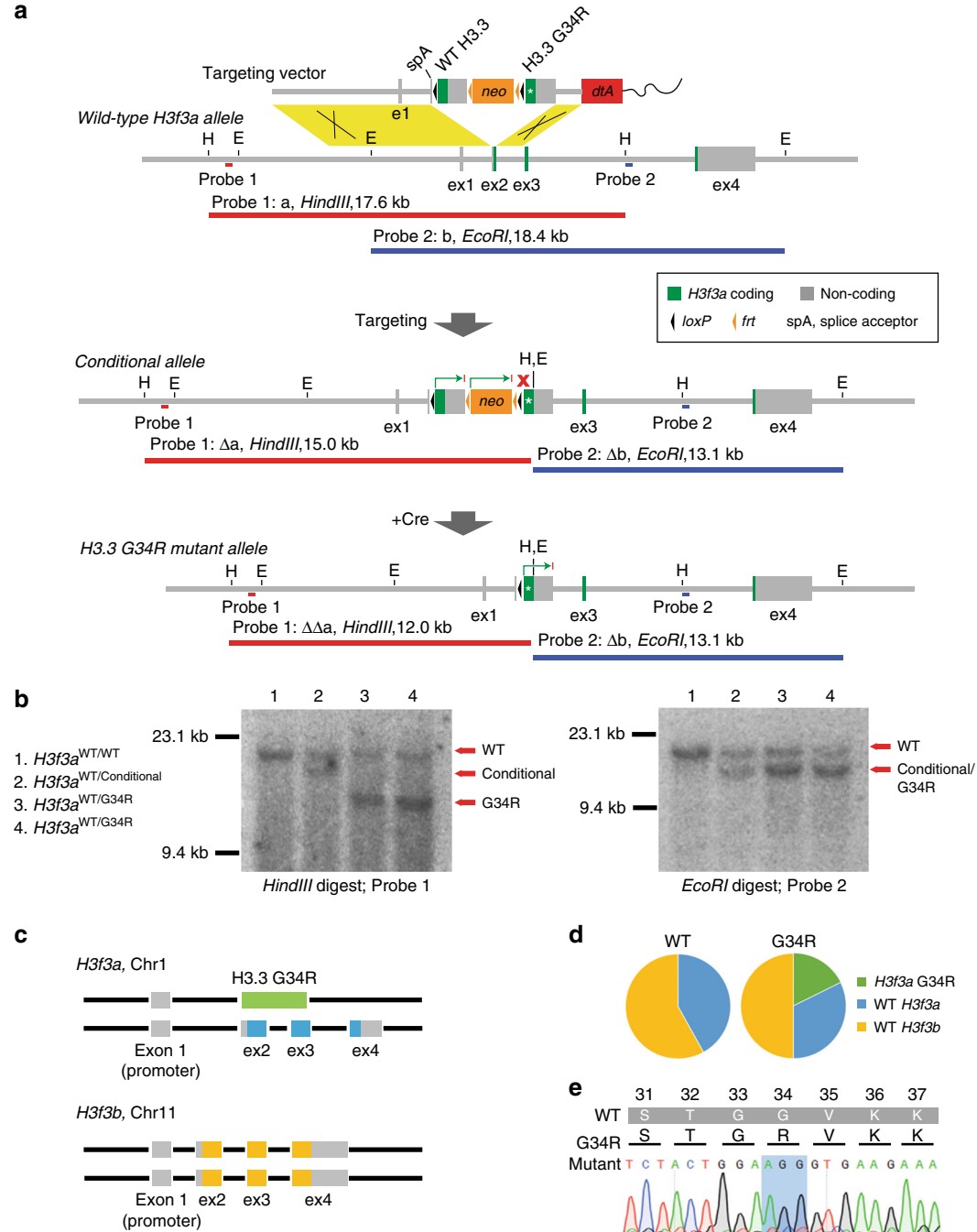

**Fig. 1** Creation and characterisation of mouse ES cells with a single copy H3.3 G34R mutation. **a** Targeting strategy for the expression of an H3.3 G34R mutant transgene from the endogenous *H3f3a* promoter. Location of Southern blot probes and expected band sizes are shown. **b** Southern blots of clones which are homozygous for WT *H3f3a*; heterozygous for the conditional allele; and heterozygous for the H3.3 G34R mutant allele. **c** Schematic representation of H3.3 genes in G34R mutants which are heterozygous WT/G34R *H3f3a* and homozygous for WT *H3f3b*. **d** Proportion of RNA-seq reads from WT and G34R cells which map to *H3f3a* G34R, WT *H3f3a* and WT *H3f3b*. **e** Sanger sequencing of cDNA from G34R cells amplified with transgene-specific primers

sequencing of cDNA amplified with transgene-specific primers (Fig. 1e).

**Increased H3K36me3 across some genes in H3.3 G34R cells.** As H3.3 G34R has previously been reported to inhibit SETD2[26], we performed H3K36me3 ChIP-seq in G34R mutants and compared this against WT cells. As H3K36me3 is predominantly associated with gene bodies and transcriptional elongation[30,31], we counted reads over H3K36me3-enriched genes and found a moderate

elevation in G34R mutants relative to WT cells (Fig. 2a, b). These results do not contradict the idea that H3.3 G34R can directly block SETD2 on the mutated histone[26], however the increase in H3K36me3 indicates that this is not the major mechanism through which H3.3 G34R exerts its effects. The gains in H3K36me3 were restricted to particular genomic regions rather than uniformly altered across the genome (Fig. 2c), which was also confirmed by ChIP-qPCR (Fig. 2d, Supplementary Fig. 2a). The increased H3K36me3 signal was not detectable by Western blots of whole cell extracts (Supplementary Fig. 2b), supporting

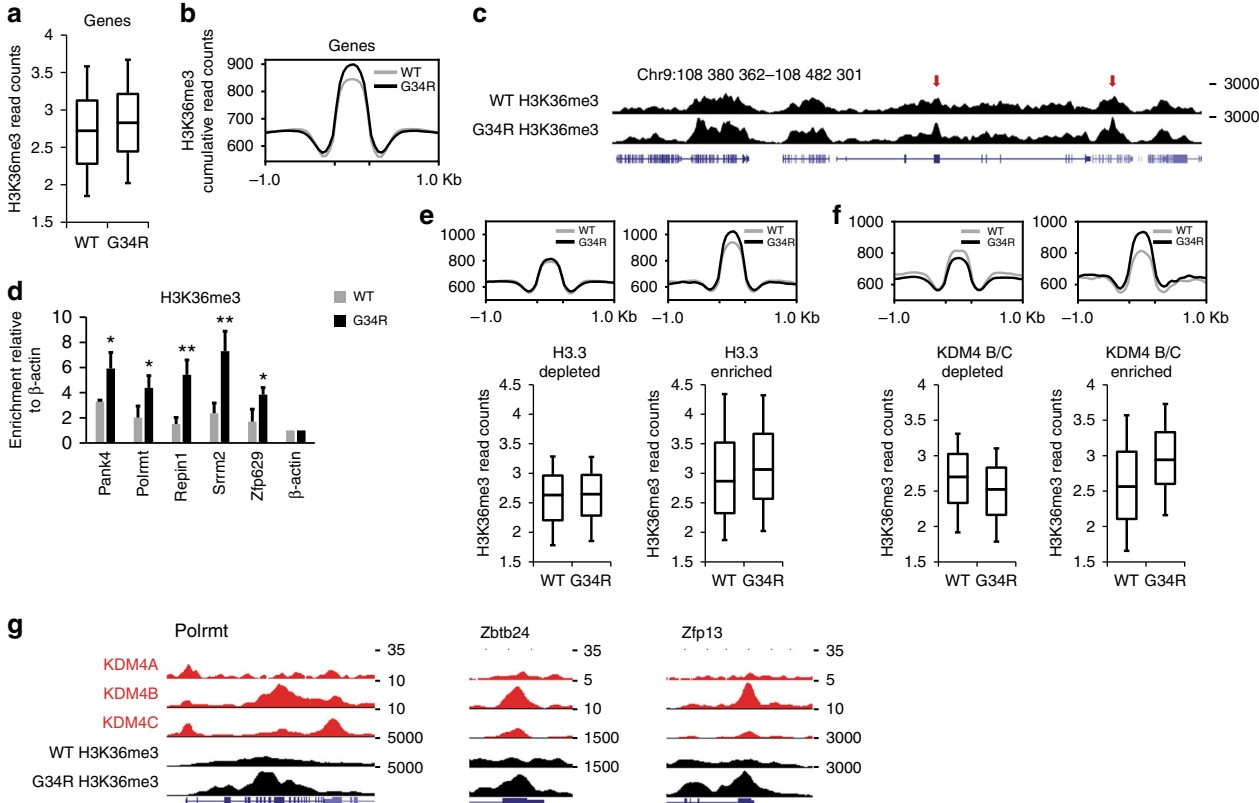

**Fig. 2** H3.3 G34R triggers gains of H3K36me3 within genes in a Kdm4-associated manner. **a** Normalised H3K36me3 ChIP-seq reads across all H3K36me3-enriched genes in WT and G34R cells. **b** Cumulative ChIP-seq read counts of H3K36me3 in WT and G34R cells. **c** Screenshot of H3K36me3 ChIP-seq in WT and G34R mutated cells. Red arrows indicate gains of H3K36me3 at discrete genomic sites. **d** H3K36me3 ChIP-qPCR in WT and G34R cells. Results are normalised for input and bars represent mean enrichment of three independent experiments, calculated relative to a negative control (β-actin promoter). Error bars represent standard deviation of three independent experiments ($n = 3$). P-values calculated using Student's T-test (*$P < 0.05$, **$P < 0.01$). **e** Cumulative and normalised H3K36me3 read counts at H3.3 depleted ($n = 7795$) and enriched ($n = 3161$) genomic regions. **f** Cumulative and normalised H3K36me3 read counts at KDM4 B/C depleted ($n = 4950$) and enriched ($n = 618$) genomic regions. **g** Screenshots of WT-KDM4 A/B/C and H3K36me3 in WT and G34R cells at three representative genes: *Polrmt*, *Zbtb24* and *Zfp13*. ChIP-seq of KDM4-A[36], -B and -C[37] were obtained from GEO (accession number GSE64252 and GSE43231, respectively). Boxes represent 25th, median and 75th percentile; whiskers represent 10th and 90th percentiles

the idea that these alterations were limited to distinct genomic sites and indicating that H3.3 G34R interferes with very specific genomic processes.

As the G34R mutation is specific to H3.3, we first sought to determine if there was a direct association between the distribution of H3.3[19] and changes in H3K36me3. The genome was binned according to H3.3 enrichment and WT/G34R H3K36me3 read counts were compared. We observed moderate gains of H3K36me3 in G34R mutants at H3.3-enriched regions (Fig. 2e), indicating the H3.3 G34R mutation might be altering H3K36me3 both directly and indirectly. This is analogous to the H3.3 K27M mutation, which alters H3K27me3 both directly and indirectly by inhibiting a histone K27 methyltransferase within the polycomb repressor 2 complex (PRC2)[26].

**Gains in H3K36me3 are associated with KDM4 binding**. We therefore reasoned that the observed gains in H3K36me3 could be due to H3.3 G34R inhibition of a K36 demethylase. The demethylation of H3K36me3 is primarily catalysed by three closely related KDM4 (histone lysine (K) *dem*ethylase) enzymes, KDM4-A, -B and -C[6,7,32]. This family of demethylases are structurally highly similar and all have dual substrate specificity towards both H3K36me3 and H3K9me3[32–34]. The crystal structures of these enzymes indicate that a small amino acid is required at the −2

position (e.g. glycine at position 34) for full catalytic activity[34,35]. We therefore asked if the gains in H3K36me3 could be due to H3.3 G34R interference with KDM4 A/B/C.

To address this, we obtained ChIP-seq of KDM4-A[36], -B and -C[37], then binned the genome according to KDM4 A/B/C enrichment and assessed H3K36me3 in WT vs G34R cells. The three demethylases had overlapping distribution profiles and the majority (2596/4207, 62%) of KDM4 binding sites were simultaneously enriched for all three KDM4 A/B/C (Supplementary Fig. 2c). The remainder fell into two major clusters; 19% (778/4207) were enriched for KDM4A alone while 15% (618/4207) were enriched for KDM4B and KDM4C (KDM4 B/C) together (Supplementary Fig. 2c). We observed an association between KDM4 enrichment and regions which gained H3K36me3 in G34R mutants, though not at sites enriched for KDM4A alone (Supplementary Fig. 2d,e). The gains in H3K36me3 were particularly prominent at KDM4 B/C-enriched regions (Fig. 2f), and examples of gene bodies which are enriched for KDM4 and gain H3K36me3 are shown (Fig. 2g). We note that KDM4 binding frequently coincided with H3.3 enrichment but we were able to isolate a number of KDM4 B/C sites with variable H3.3 content (Supplementary Fig. 2f). We observed H3.3-independent gains in H3K36me3 at these sites (Supplementary Fig. 2g), indicating the H3.3 G34R mutant can exert effects by interfering with KDM4 function.

**H3.3 G34R cells gain H3K9me3 in a KDM4-associated manner**. As KDM4 has dual (K9/K36) demethylation activity, this model predicts H3K9me3 profiles would also be altered in H3.3 G34R mutants. To test this, we performed H3K9me3 ChIP-seq in WT and G34R cells and isolated genic regions which were enriched for H3K9me3 in either cell type. Similar to H3K36me3, we observed moderate gains of H3K9me3 across all of these regions (Fig. 3a). We next asked if the gains of H3K9me3 in the G34R mutants could be attributed to disruption of KDM4. We binned these regions according to KDM4 A/B/C enrichment (the three enzymes were inseparable at these sites, Supplementary Fig. 3a) and found that, similar to H3K36me3, high levels of KDM4 binding predicted gains of H3K9me3 in G34R cells (Fig. 3b). Together, these results demonstrate that expression of H3.3 G34R causes gains of both H3K9me3 and H3K36me3 in a KDM4-associated manner, and supports the idea that H3.3 G34R is inhibiting the KDM4 demethylases.

**H3.3 G34R mutants are similar to KDM4 A/B/C triple knockouts**. If H3.3 G34R is able to directly inhibit KDM4, we would expect to see similarities between the G34R mutants and KDM4 knockout cells. This was examined by comparing H3K9me3/H3K36me3 chromatin profiles in a KDM4 A/B/C triple-knockout (KDM4-tKO)[36] to the H3.3 G34R profiles. We identified regions which gained H3K9me3 (>1.5-fold) in KDM4-tKO cells and observed comparable gains at these sites in the H3.3 G34R mutant (Fig. 3c). Genomic regions where H3K9me3 were unchanged (<1-fold difference) in KDM4-tKO cells also remained unchanged in the H3.3 G34R cells (Supplementary Fig. 3b). A similar relationship was observed with H3K36me3; the gains of H3K36me3 in KDM4-tKO cells were mirrored in the G34R cells (Fig. 3d) while sites where H3K36me3 were stable in KDM4-tKO also remained constant in G34R mutants (Supplementary Fig. 3c). Examples of KDM4-enriched sites which gain both H3K9me3 and H3K36me3 in KDM4-tKO and G34R mutants are shown (Fig. 3e), and gains of H3K9me3 at some sites were confirmed by ChIP-qPCR (Fig. 3f, Supplementary Fig. 3d).

As the KDM4 family are known transcriptional regulators, we next asked if the transcriptional changes in G34R cells resembled the KDM4-tKO mutants. We obtained a list of genes which were identified as either down- or up-regulated by microarray in KDM4-tKO cells[36]. The microarray gene identifiers were cross-referenced against the UCSC database to retrieve mm9 gene coordinates; 944/1005 (94%) of KDM4-tKO downregulated genes and 1099/1107 (99%) of KDM4-tKO upregulated genes were identified. We assessed WT and G34R RNA-seq data and observed clear similarities between the G34R and KDM4-tKO mutants. Genes which were downregulated in the KDM4-tKO mutants tended to be downregulated in G34R cells relative to WT expression and compared against a sample of random genes (Fig. 3g,h). One example of a gene which is downregulated in both G34R and KDM4-tKO is shown (Fig. 3i). A similar profile was observed in the KDM4-tKO upregulated cohort, as these genes also tended to be upregulated in G34R cells (Fig. 3j–l). Transcriptional changes for a number of genes were confirmed by qRT-PCR (Supplementary Fig. 3e). Combined, these results demonstrate that expression of a single copy of H3.3 G34R is sufficient to trigger chromatin and transcriptional changes which are highly similar to those observed in KDM4-tKO mutants, strongly suggesting that this is a major pathway through which H3.3 G34R exerts its effects.

**H3.3 G34R binds and inhibits KDM4**. As the H3.3 G34R mutant closely resembles a KDM4-tKO, this suggests a dominant-negative model where H3.3 G34R binds and inhibits KDM4. To test if the KDM4 demethylases preferentially bind to H3.3 G34R, we co-expressed Flag/biotin-tagged KDM4 A/B/C with either WT- or G34R-HA-tagged H3.3. We then immuno-precipitated KDM4 A/B/C and blotted for HA-H3.3 binding. We consistently detected higher levels of H3.3 G34R compared to WT H3.3 (Fig. 4a), demonstrating that this single substitution is sufficient to increase the binding preference between full-length H3.3 G34R and KDM4 in the cellular context. The preferential interaction between H3.3 G34R and KDM4 were confirmed across three independent experiments (Supplementary Fig. 4a). Equal expression of WT H3.3, H3.3 G34R and KDM4 A/B/C (Supplementary Fig. 4b), as well as equal precipitation of KDM4 (anti-Flag) (Supplementary Fig. 4c–e), were confirmed by Western blots. To test if this preferential interaction persisted when the K36 residue is methylated, we used an in vitro binding assay between recombinant KDM4 and WT H3.3 K36me3 and H3.3 G34R K36me3 peptides (Supplementary Fig. 4f). As expected, we found that KDM4-A and -C interacted with both peptides, but we again detected stronger binding of the H3.3 G34R peptide with KDM4 compared to WT H3.3 (Supplementary Fig. 4g, h).

To determine if H3.3 G34R altered the distribution of KDM4 on chromatin, we expressed Flag-Biotin-tagged KDM4B in WT and H3.3 G34R cells, and assessed binding by ChIP-qPCR. We selected a gene (Zfp358) which had two peaks of KDM4B binding in WT cells; one centred on the promoter with relatively low levels of H3.3, and a second peak within the gene body with high levels of H3.3 (Fig. 4b). Using qPCR primers tiled across this gene, we found that KDM4B binding was significantly reduced at the promoter but retained at the high H3.3, intragenic site (Fig. 4b). These findings support the idea that H3.3 G34R preferentially binds to KDM4 and is able to recruit this demethylase from endogenous binding sites.

As sites that are enriched for H3.3 also show increased H3K36me3 (Fig. 2e), this suggests that H3.3 G34R may also inhibit the catalytic activity of KDM4. Structural studies indicate that the H3 G34 residue is important for the correct positioning of KDM4 relative to H3K36me3[34,35]. KDM4 has also been shown to have a preference for a small amino acid in the −2 position relative to H3K9me3 and H3K36me3 targets (e.g. A7 and G34), and an A7R substitution interfered with KDM4A demethylation of H3K9me3[34]. To determine if the G34R substitution inhibited KDM4 demethylation of K36me3, recombinant KDM4-A, -B and -C were combined with either a H3.3 K36me3 or an H3.3 G34R K36me3 peptide (Supplementary Fig. 4f) and results were analysed by mass spectrometry. KDM4-A, -B and -C were able to demethylate WT H3.3 K36me3 (Fig. 4c, left panel) while the G34R mutation substantially inhibited the demethylase activity of all three enzymes (Fig. 4c, right panel). These results support a model where H3.3 G34R preferentially binds to KDM4 and inhibits its activity, analogous to the relationship between H3.3 K27M and PRC2[26].

## Discussion
Histone point mutations are a common feature of certain cancers including chondroblastomas (H3.3 K36M)[24], head and neck squamous cell carcinomas (H3 K36M)[25], giant cell bone tumours (H3.3 G34W/L)[24] and paediatric gliomas (H3.3 K27M, H3.3 G34R/V)[22,23]. The lysine to methionine substitutions are thought to bind and inhibit lysine methyltransferases to trigger genome-wide changes in chromatin states[26,27]. The mechanism which underlies the G34 substitutions have been less clear. Early studies indicated that an H3.3 G34R substitution prevented SETD2-mediated tri-methylation of the neighbouring K36 residue but this effect is limited to the mutated histone[26], unlike the relationship between K27M/PRC2 and K36M/SETD2 which are able to act in trans across the genome[26,27].

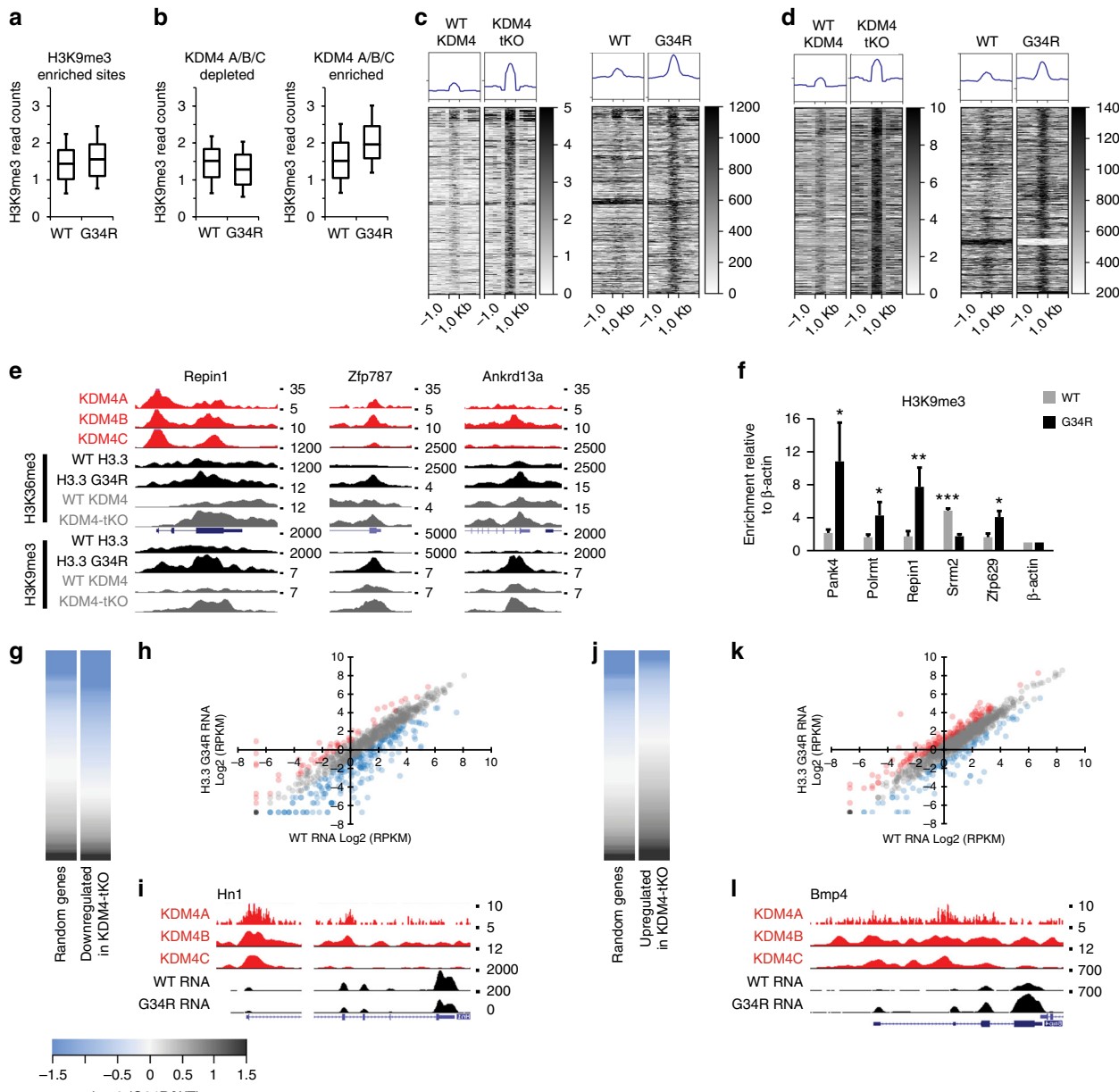

**Fig. 3** H3.3 G34R triggers gains in H3K9me3 and genocopies KDM4 A/B/C triple-knockout. **a** Normalised H3K9me3 ChIP-seq reads across all H3K9me3-enriched genes in WT and G34R cells. **b** Normalised H3K9me3 read counts at KDM4 A/B/C depleted ($n = 2953$) and enriched ($n = 4761$) genomic regions. **c**, **d** Heatmaps of **c** H3K9me3 and **d** H3K36me3 ChIP-seq reads at sites which gain (>1.5-fold) H3K9me3 ($n = 927$) or H3K36me3 ($n = 439$) in KDM4-tKO mutants (±1 kb) with profiles of corresponding regions in WT and G34R cells. **e** Representative screenshots of genes (*Repin1, Zfp787, Ankrd13a*) enriched for KDM4 which gain H3K9me3 and H3K36me3 in both KDM4-tKO cells and G34R mutants. H3K9me3 and H3K36me3 ChIP-seq in KDM4-tKO[36] were obtained from GEO (accession number GSE64252). **f** H3K9me3 ChIP-qPCR in WT and G34R cells. Results are normalised for input and bars represent mean enrichment of three independent experiments, calculated relative to a negative control (β-actin promoter). Error bars represent standard deviation of three independent experiments ($n = 3$). *P*-values calculated using Student's *T*-test (\**P* < 0.05, \*\**P* < 0.01, \*\*\**P* < 0.0001). **g** Heatmap of $\log_2$ (G34R/WT RPKM) of genes which were downregulated in KDM4-tKO ($n = 933$) compared against a random selection ($n = 933$) of genes. **h** $\log_2$ (RPKM) of WT vs G34R RNA-seq of genes downregulated in KDM4-tKO ($n = 933$). **i** Representative screenshot of a gene which is downregulated in KDM4-tKO. **j** Heatmap of $\log_2$ (G34R/WT RPKM) of genes which were upregulated in KDM4-tKO ($n = 1099$) compared against a random selection ($n = 1099$) of genes. **k** $\log_2$ (RPKM) of WT vs G34R RNA-seq of genes upregulated in KDM4-tKO ($n = 1099$). **l** Representative screenshot of a gene which is upregulated in KDM4-tKO. Red and blue circles indicate genes which are >2-fold up or downregulated in G34R relative to WT. Boxes represent 25th, median and 75th percentile; whiskers represent 10th and 90th percentiles

We sought to characterise and identify the chromatin modifications and pathways which are most affected by the H3.3 G34R mutation. In paediatric gliomas, the H3.3 G34R substitution is always found in conjunction with mutations in an H3.3 chaperone, ATRX[22]. The double mutation in H3.3 and its chaperone, means that it has not been possible to accurately

determine the chromatin alterations which are directly attributable to H3.3 G34R from analyses of patient samples. We addressed this by recreating the *H3f3a* G34R single-copy mutation in mouse ES cells on an otherwise WT background. The initial analysis of this cell line showed that there were localised increases in H3K36me3 at specific regions across the genome.

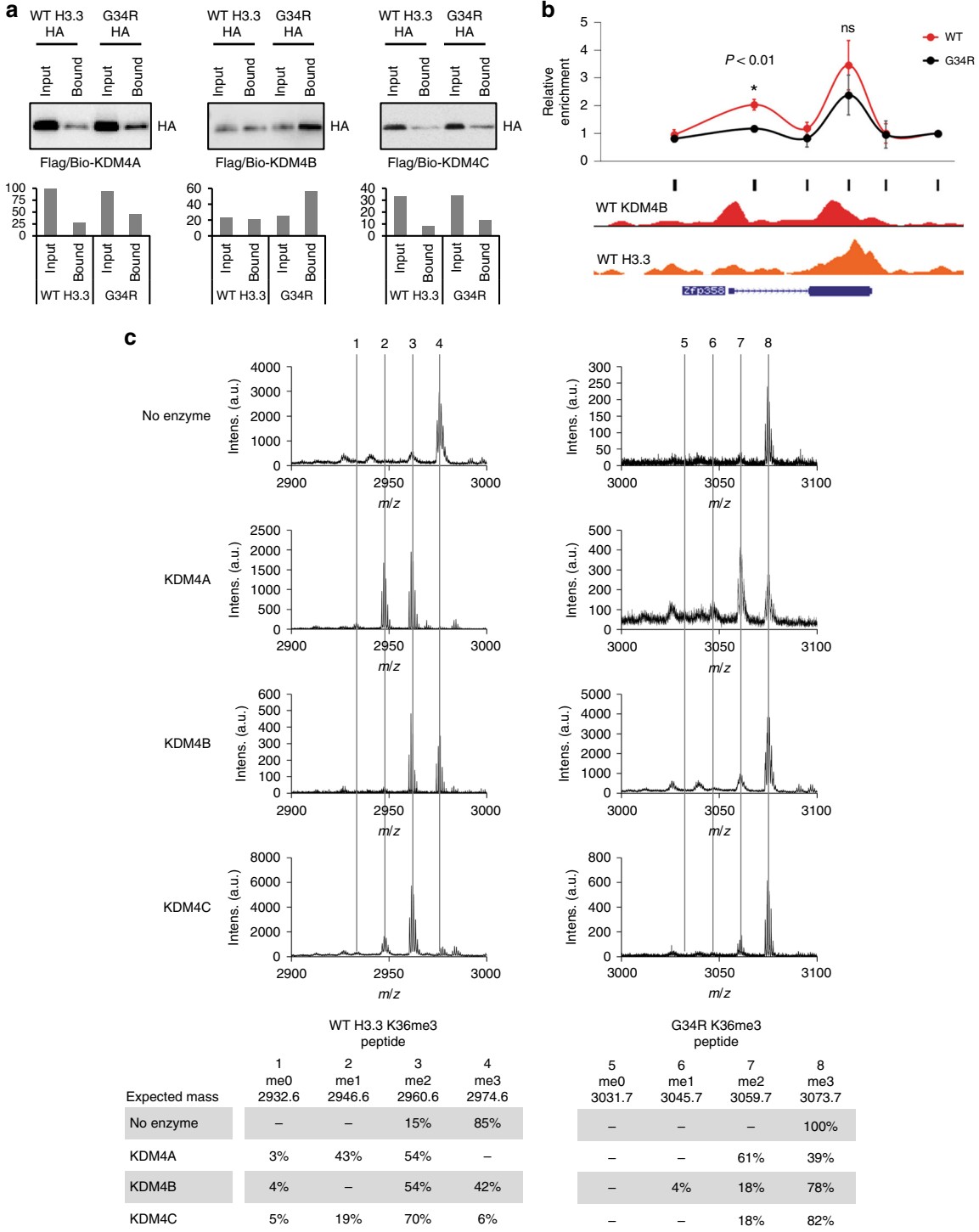

**Fig. 4** H3.3 G34R preferentially binds to KDM4 and inhibits its demethylase activity. **a** Protein pulldowns of KDM4-A, -B and -C, immunoblotted with anti-HA for detection of HA-tagged WT H3.3 and H3.3 G34R. Representative blots are shown and quantitation values are shown. **b** ChIP-qPCR of KDM4B in WT and G34R cells tiled across a representative gene, *Zfp358*. Results are normalised for input and values represent mean enrichment of three independent experiments, calculated relative to a negative control (*Zfp358*, primers 6). Data are mean average and error bars are standard deviations of three independent experiments (*n* = 3). *P*-values calculated using Student's *T*-test (**P* < 0.01). Black bars indicate primer locations. ChIP-seq profiles of KDM4B and H3.3 in WT cells across the gene are shown. **c** Mass spectrometry analysis of in vitro demethylase assays. KDM4-A, -B and -C were combined with either WT H3.3 K36me3 peptide (left panel) or a G34R K36me3 peptide (right panel). Dashed lines indicate expected masses for K36 me0, me1, me2 and me3

These observations do not directly contradict the previous reports that H3.3 G34R prevents SETD2-mediated trimethylation of H3.3 G34R K36, however our findings strongly suggest the H3.3 G34R is able to alter the genome *in trans* through a separate chromatin pathway.

We reasoned that these chromatin changes may be due to inhibition of the H3K36me3 demethylases, KDM4 A/B/C[6,7], which reportedly prefer a small amino acid (e.g. glycine) at position 34 of H3/H3.3[34,35]. As these demethylases have dual activity towards both H3K36me3 and H3K9me3, we profiled H3K9me3 and found this modification was also altered in the G34R mutant. Furthermore, the chromatin and transcriptional profiles of the G34R mutant were highly similar to those observed in a KDM4-tKO. Therefore, expression of a single copy of H3.3 G34R at endogenous levels is sufficient to alter two independent histone modifications in a manner similar to a KDM4-tKO, strongly suggesting that H3.3 G34R is interfering with the KDM4 demethylases.

Consistent with this, we were able to demonstrate that KDM4 shows binding preference for H3.3 G34R compared to WT H3.3, and this preference is sufficient to alter the distribution profile of KDM4B in an H3.3-dependent manner. It is important to note that a substantial fraction of H3.3 is incorporated into genomic repeats and our current assays were not able to fully assess the H3.3 G34R-mediated disruptions of KDM4 within these repeats. Further studies into H3.3 G34R and the other H3.3-specific oncohistones (e.g. H3.3 K27M, H3.3 K36M) should attempt to address this critical issue. Nonetheless, we were able to demonstrate that H3.3 G34R preferentially binds to KDM4 and also inhibits demethylase activity towards H3.3 G34R K36me3. The simultaneous binding and inhibition of KDM4 by H3.3 G34R, is reminiscent of the interactions between K27M/PRC2 or K36M/SETD2, and accounts for the similarities between the G34R and KDM4-tKO mutants. These findings are also consistent with observations of increased H3K36me3 in H3.3 G34V brain tumours[22,38], suggesting that the G34V mutations may also inhibit KDM4. We do not exclude the possibility that H3.3 G34R may affect additional chromatin pathways, but we nonetheless identify KDM4 as the major pathway which is disrupted by this mutation.

Many studies have indicated that defects in KDM4 contribute to oncogenesis[7], including neomorphic mutations in IDH1/2 which produce an oncometabolite[9–11] that inhibits 2-OG-dependent dioxygenase demethylases, including the KDM4 family of demethylases[12,13]. The IDH1/2 mutations are relatively rare in paediatric gliomas but occur at high frequency in adult lower grade gliomas. Mutations in IDH1/2 are a defining feature of diffuse astrocytomas and are almost always found in conjunction with inactivating mutations in ATRX and TP53[14–17]. This mutation signature is identical to H3.3 G34R mutated paediatric gliomas, which are similarly mutated for ATRX and TP53[22]. This raises the possibility that mutations in H3.3 G34R and IDH1/2 act through parallel pathways, and inhibition of KDM4 may be a common feature in ATRX mutated cancers.

It is now clear that mutations in chromatin pathways are a common feature in many cancers, and the majority of mutations trigger broad changes in chromatin states. We have demonstrated that H3.3 G34R also fits this general model and outline a mechanism which enables this point mutation to alter chromatin profiles across the genome. These findings broaden the framework for understanding how histone mutations may exert their effects and suggest that, much like histone modifications, these mutations act through interactions with epigenetic readers, writers and erasers.

## Methods

**Cell culture.** Mouse ES cells were cultured in Dulbecco Modified Eagle Medium supplemented with 15% heat-inactivated foetal calf serum, $10^3$ units/ml leukaemia inhibitory factor (Merck) and 0.1 mM β-mercaptoethanol. 'WT' cells indicate unmodified ES cells ($H3f3a^{+/+}$; $H3f3b^{+/+}$). 'G34R' cells refer to cells with a single $H3f3a$ G34R mutation ($H3f3a^{+/G34R}$; $H3f3b^{+/+}$).

**Generation of H3.3 G34R mouse ES cell line.** The single-copy H3.3 G34R mutant ES cells lines used in these studies were created using a previously described conditional allelic replacement strategy[29]. The targeting vector was identical with the previous study, except that the yellow fluorescent protein coding sequence (CDS)[29] was replaced with H3.3 G34R CDS. On exposure of targeted cells to Cre-recombinase, the WT H3.3 minigene CDS was excised, and the mutant G34R minigene CDS was brought under control of the endogenous $H3f3a$ promoter. Mutant clones were selected by restoration of G418 sensitivity (as the *neo* cassette is excised together with the WT minigene), and confirmed by Southern blot. The presence of mutant H3.3 G34R RNA transcript was validated by reverse transcription-PCR amplification of mRNA followed by DNA sequencing analysis of the PCR products using transgene-specific sequencing primers (Supplementary Table 1).

**Southern blotting.** Southern Blots were performed as previously described[39]. In brief, cells were lysed with 0.5 ml of lysis buffer (50 mM Tris–HCl pH 8.0, 100 mM NaCl, 100 mM EDTA, 1% SDS). DNA was precipitated with ammonium acetate and isopropanol, and washed with 70% ethanol. DNA was resuspended in 50 μl of TE buffer. Genomic DNA was digested with a 5-fold excess of restriction enzyme and 5 μg was loaded. Blots were performed with Hybond XL membranes according to manufacturer's protocols (GE Healthcare), except that the pre- and hybridisation solution was 5× SSPE, 5× Denhardt's solution, 1% SDS. All gels were depurinated and washed twice with 0.2× SSC, 0.5% SDS. PCR primers for probe synthesis are shown in Supplementary Table 1. Full blots of Southern blot analyses in Fig. 1b are shown in Supplementary Fig. 5.

**Antibodies.** Antibodies used were directed against H3 (Abcam ab1791), H4 (Merck Millipore), H3.3 (Merck Millipore 09-838), H3K36me3 (Abcam ab9050), H3K9me3 (Abcam ab8898), H4K20me3 (Abcam ab9053), Flag (Sigma Aldrich), HA (Roche) and Actin (Santa Cruz Biotechnology).

**Chromatin immunoprecipitation.** Five million cells were fixed in growth media with 0.4% formaldehyde for 10 min and quenched with 125 mM glycine. Chromatin was released by sequential lysis with cells lysis buffer (10 mM Tris pH 8, 10 mM NaCl, 0.2% Igepal) and nuclear lysis buffer (50 mM Tris pH 8, 10 mM EDTA, 1% SDS). Chromatin was sheared to ≈300 bp with 12 rounds of sonication (30 s on, 30 s off) on a Bioruptor (Diagenode) and incubated overnight at 4 °C with either 5 μg H3K9me3 antibody (Abcam ab8898) or 10 μg H3K36me3 antibody (Abcam ab9050). Samples were immunoprecipitated with Protein A Agarose beads (Sigma-Aldrich, 05015979001) and washed sequentially with low salt (20 mM Tris pH 8, 2 mM EDTA, 50 mM NaCl, 1% Triton X-100, 0.1% SDS), high salt (20 mM Tris pH 8, 2 mM EDTA, 500 mM NaCl, 1% Triton X-100, 0.01% SDS) and LiCl buffers (10 mM Tris pH 8, 1 mM EDTA, 0.25 M LiCl, 1% Igepal, 1% sodium deoxycholate). Chromatin was eluted with elution buffer (1% SDS, 100 mM NaHCO₃), decrosslinked overnight at 65 °C, incubated with Proteinase K, phenol chloroform extracted and ethanol precipitated.

For KDM4B ChIP, mouse WT and G34R ES cells were transfected with Flag/Biotin-tagged KDM4B using Lipofectamine 2000 Transfection Reagent (Thermo Scientific). After 24 h, transfected cells were cross-linked first with 2 mM EGS (ethylene glycol bis(succinimidyl succinate)) (Pierce) for 45 min then subsequently with 1% paraformaldehyde for 15 min. Excess formaldehyde was quenched with glycine at a final concentration of 0.25 M. Cell were then washed, lysed and subjected to sonication as described above. Resulted chromatin was diluted in dilution buffer and immunoprecipitated with 25 μl anti-Flag-bound magnetic beads (Sigma Aldrich) at 4 °C overnight. The immunoprecipitated chromatin was washed, eluted, decross-linked and purified as described above.

**ChIP sequencing.** Sample concentrations were determined by Qubit (Thermo-Fisher Scientific) and 20 ng of DNA was used as starting material. ChIP libraries were prepared with Nugen Ovation Ultralow System V2 (Nugen protocol M01379v1, 2014) with 10 cycles of amplification. Libraries quality was assessed by Qubit, Bioanalyzer (Agilent) and qPCR, and a single equimolar pool was made based on size-adjusted qPCR quantitation. Following denaturation, 12 pM of library pools were used for cBot hybridisation and cluster generation (Illumina Protocol 15006165 v02, Feb 2016), and samples were sequenced on an Illumina HiSeq 1500 rapid mode (50 bp SR sequencing, Illumina Protocol 15035788 Rev D, Apr 2014).

**Real-time PCR.** ChIP samples were resuspended in 1 ml. 5 μl of ChIP material was combined with 0.5 μM of primers and FastStart DNA Master SYBR Green (Roche, 03003230001) in a 15 μl reaction and amplified in a LightCycler (Roche). Samples

were normalised for input and compared against a negative control point (β-actin promoter). Results and error bars are the mean average and standard deviations of three independent experiments. *P*-values were calculated using Student's *T*-test. See Supplementary Table 1 for a full list of primers used in this study.

**ChIP-seq analysis.** ChIP-seq of H3K36me3 and H3K9me3 from WT and H3.3 G34R mutants were aligned to the mouse genome (mm9) with Bowtie (v1.0.0)[40] with default settings except (m −5) to allow restricted multi-mapping to short repeats. Data were imported into Seqmonk (v 0.32.1) with PCR duplicates excluded. Reads were counted across 200 bp genomic bins and samples were normalised for total reads (per million reads). WT and H3.3 G34R datasets were normalised against each other using the 'match distribution quantitation' tool on Seqmonk.

We selected bins which overlapped UCSC annotated genes and only analysed sites which were enriched for H3K36me3/H3K9me3 (>95th percentile) in either WT or G34R cells. All windows within 100 bp were merged using MergeBed (−d 100 −o mean) function in BedTools (v 2.20.1). Merged regions >500 bp and regions with extreme read counts (top 0.01%), typically associated with poorly annotated genomic regions, were excluded from further analysis. A total of 31,593 WT/G34R H3K36me3-enriched sites and 32,502 WT/G34R H3K9me3-enriched sites were used for downstream analyses.

H3K36me3- or H3K9me3-enriched regions which exceeded the 90th percentile for H3.3 binding or the 75th percentile for KDM4 A/B/C binding within each dataset were determined to be 'enriched'. Regions which fell below the 25th percentile were defined as 'depleted'. Comparison plots show normalised H3K9me3 or H3K36me3 ChIP-seq reads from WT and G34R cells under regions deemed to be enriched or depleted. Boxes represent the 25th, median and 75th percentiles; whiskers represent the 10th and 90th percentiles. Venn diagrams show overlaps between regions deemed to be enriched for binding, with depletion was used as exclusion criteria. Cumulative read profiles were generated in DeepTools using computeMatrix (v 2.5.0.0) and plotProfile (v 2.5.0.0) on Galaxy. Regions were fitted to 500 bp with an additional 1 kb up and downstream; blacklisted[41,42] regions were excluded from analysis.

See Supplementary Table 2 for a full list of GEO datasets used in this study.

**Comparison with KDM4-tKO datasets.** ChIP-seq of H3K9me3 and H3K36me3 from WT-KDM4 and KDM4-tKO cells were obtained from ref. [36] (Supplementary Table 2) and treated as described above. A total of 22,237 WT-KDM4/KDM4-tKO H3K9me3-enriched sites and 27,951 H3K36me3-enriched sites were identified. To ensure that the WT/G34R cohort demonstrated enrichment for H3K9me3 or H3K36me3 at these sites, a further cut-off was established based on percentile values of WT cells. H3K9me3 values had to exceed the 90th percentile in either WT or G34R cells; yielding a total of 4237 regions which are enriched for H3K9me3 in WT-KDM4/KDM4-tKO and WT/G34R cells. H3K36me3 values had to exceed the 75th percentile in either WT or G34R cells yielding a total of 11,454 regions.

Regions where H3K9me3 or H3K36me3 was 1.5-fold higher in KDM4-tKO relative to WT-KDM4 were deemed to have gained these modifications in KDM4-tKO cells. Regions where KDM4-tKO/WT-KDM4 ratios were between 0.8 and 1 were deemed to be 'unchanged'. Heatmaps were created using DeepTools computeMatrix (v 2.5.0.0) and plotHeatmap (v 2.5.0.0) in Galaxy. Regions where H3K9me3 and H3K36me3 were gained or unchanged in KDM4-tKO cells were scaled to fit in 500 bp with an additional 1 kb up and downstream; WT-KDM/KDM4-tKO and WT/G34R H3K9me3 and H3K36me3 ChIP-seq were used as score files.

**RNA extraction and sequencing.** RNA was extracted from 5 million cells with TRI Reagent (Sigma, 93289) according to manufacturer's instructions. Samples quality was assessed using a Bioanalyzer and quantified by Qubit. All samples were processed starting with 500 ng of total RNA according to the Illumina Stranded Total RNA protocol (15031048 Rev C, Sept 2012) except that 12 cycles of amplification was used to minimise amplification artefacts. Libraries were quantitated by Qubit and qPCR, and an equimolar pool were made based upon qPCR results. Following denaturation, 200 pM of the library pool was clustered in one lane of a HiSeq 3000 (Illumina Protocol 15006165 v02, Jan 2016) and 50 bp SR sequencing was carried out (Illumina Protocol 15066493 Rev A, February 2015).

**RNA-seq analysis.** Stranded 50 bp single-end RNA-seq reads were aligned to mouse (mm9) with STAR (v.2.4.2a)[43] using default parameters. Primary alignments were imported into Seqmonk as single-end RNA-seq. The RNA-seq quantitation pipeline was used to estimate RPKM read counts with the following parameters: mRNA as transcript feature, strand-specific, merge transcript isoforms, apply transcript length correction and exclude probes with no counts. A total of 26,127 mRNA probes were quantitated.

A list of genes which were either downregulated or upregulated in KDM4-tKO cells as determined by microarray, was obtained from ref. [36]. The UCSC Table Browser was used to extract mm9 coordinates based on gene names. A total of 944 (of 1005) downregulated and 1107 (of 1121) upregulated genes had mapped coordinates on the mm9 genome. Genes which were up/downregulated in KDM4-tKO cells were overlapped with RNA quantitations for WT/G34R cells. Genes which failed to overlap were excluded from further analyses, yielding a total of 933

KDM4-tKO downregulated genes and 1099 upregulated genes with matched WT and G34R RNA-seq data. $\log_2$ (G34R/WT) RPKM values were calculated for KDM4-tKO up and downregulated genes and sorted by ascending values. Results were plotted in R Studio using the gplots package[44].

A full list of genes, which were up- or down-regulated in G34R cells relative to WT, are provided as Supplementary Data 1.

**Identification of H3.3 reads.** RNA-seq reads from WT and G34R cells were aligned to a custom library, consisting of the CDS for H3f3a G34R transgene, *H3f3a* (NM_008210.5) and *H3f3b* (NM_008211.3), using Bowtie (v1.0.0) (2) with default settings. Reads which spanned codon 34 were extracted and duplicates were removed. A total of 62 and 68 informative reads were identified in WT cells and G34R cells, respectively. Pie charts show the proportion of reads which were identified as originating from H3.3 G34R, WT H3f3a or WT H3f3b.

**RT-qPCR for expression analysis.** RNA extracted from WT and G34R cells were quantitated by Nanodrop and 1 μg of total RNA was reverse transcribed (ThermoFisher Scientific, 4368813) according to manufacturer's instructions. 20 ng of cDNA was combined with 0.5 μM of primers and FastStart DNA Master SYBR Green (Roche, 03003230001) in a 15 μl reaction and amplified in a LightCycler (Roche). Expression was calculated relative to a gene (*Ddx28*) which was unchanged between WT and G34R cells. Results and error bars are the mean average and standard deviations of three independent experiments. *P*-values were calculated using Student's *T*-test. Primers are shown in Supplementary Table 1.

**Immunoprecipitation.** Mouse ES cells were transfected with Flag/Bio-tagged KDM4 A/B/C, and HA-tagged WT H3.3 or H3.3 G34R using Lipofectamine 2000 Transfection Reagent (Thermo Scientific). After 24 h, transfected cells were then lysed in cold RIPA buffer (50 mM Tris–HCl pH 7.5, 1 mM EDTA, 150 mM NaCl, 1% NP40, 0.5% sodium deoxycholate, 0.05% SDS and protease inhibitors) and sonicated. After centrifugation, the supernatant was collected and subjected to immunoprecipitation overnight using streptavidin magnetic beads and anti-FLAG antibody (Sigma, M2 mouse monoclonal; used at 1/1000). Beads were washed with RIPA buffer three times. Immunoprecipitants were eluted by addition of 2× SDS PAGE sample buffer containing beta mercaptoethanol and boiled at 95 °C for 5 min. The eluates were subjected to SDS/PAGE gel electrophoresis and Western blot with an anti-HA antibody (Merck, 3F10 rat monoclonal; used at 1/5000). Bands were quantitated with ImageJ software and normalised to input. Results and error bars are the mean average and standard deviations of three independent experiments. *P*-values were calculated using Student's *T*-test. Full blots of Western blot analyses in Fig. 4a are shown in Supplementary Fig. 6.

**Histone demethylase assay.** 250 ng of either H3.3 WT K36me3 or H3.3 G34R K36me3 peptides (GLC Biochem, China) were incubated with 50 nM KDM4 A/B/C recombinant protein in reaction buffer (50 μl) containing 50 mM Tris–HCl pH 7.5, 0.02% Triton X-100, 200 μM 2-OG, 200 μM ascorbate, 50 μM $(NH_4)_2Fe$ $(SO_4)_2 \cdot 6(H_2O)$, 1 mM TCEP for 2 h at room temperature. Samples were co-spotted onto an MTP anchorChip 800/384 TF MALDI target plate with Matrix solution of 10 mg/ml a-cyano-4-hydroxycinnamic acid (Laser BioLabs, Sophia-Antipolis, France) in 50% acetonitrile 0.1% TFA. Samples were analysed on a Bruker Daltonics (Bremen, Germany) ULTRAFLEX MALDI TOF/TOF in reflector mode with an *m/z* range of 1200–3500 Da, using Smartbeam parameter set 4, and detector gain 2.5× for 1000 shots. The data was processed using Flexanalysis (Version 3.4, build 50). The spectra were externally calibrated against a peptide mix including angiotensin (1296 *m/z*), Glu-Fibrinopeptide B (1570 *m/z*), ACTH[1–17] (2093 *m/z*), ACTH[18–39] (2465 *m/z*) and ACTH[7–38] (3657 *m/z*) which were spotted on adjacent calibration wells.

**In vitro binding assays.** 40 ng of biotinylated H3.3 K36me3 and H3.3 G34R K36me3 peptides (GLC Biochem) were bound to 20 μl of streptavidin magnetic beads (Thermo Scientific) in 200 μl of PBS for 5 h at 4 °C. Beads were washed twice with PBS then blocked with 1% BSA in PBS. Excess BSA was removed with a final PBS wash. 200 ng of recombinant Flag-tagged KDM4A or KDM4C (Active Motif) were incubated with peptide-bound beads in PBS overnight at 4 °C. A fraction of the input protein and the supernatant after binding to beads (flowthrough) was retained for gel analysis. Beads were washed with PBS. Input, flowthrough and bead fractions were run on a 10% SDS-PAGE gel and analysed by western blotting with an anti-FLAG antibody (Sigma, M2 mouse monoclonal).

**Data availability.** Datasets used in this study are listed in Supplementary Table 2. The ChIP and RNA sequencing datasets generated for this study are available on the Gene Expression Omnibus (GEO) database under the accession number GSE106205.

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

## Acknowledgements

This work was supported by the Australia Research Council (ARC; Future Fellowship Award to L.H.W.), the Australian National Health and Medical Research Council (L.H. W. Project Grant), Cure Brain Cancer Australia (L.H.W. and H.P.J.V. Project Grant) and The Isabella and Marcus Paediatric Brainstem Tumour Fund (L.H.W. Project Grant). In memory of Mei Ling Voon.

## Author contributions

H.P.J.V. and L.H.W. designed the experiments and prepared the manuscript. H.P.J.V., M. U., W.L., L.H. and L.H.W. performed most of the experiments. R.H.P.L. and D.L.S. advised on the interaction and demethylase assays. D.L.S. performed the mass spectro-metry. P.P.D. advised on KDM4 analyses and provided KDM4 expression vectors. J.R.M. and L.H.W. designed and created the H3.3 G34R cell lines and performed the Southern Blots.

## Additional information

**Competing interests:** The authors declare no competing interests.

