## [Peer Review File · Nature Communications]

Reviewers' comments:

Reviewer #1 (Remarks to the Author):

This study describes the generation and characterisation of a mouse ES cell line that expresses the H3.3 G34R mutant from the endogenous locus. Since the H3.3 G34R mutation has been linked to the H3K36me3 specific methyltransferase SETD2, the authors performed H3K36me3 ChIP-seq in WT and H3.3 G34R expressing cells. This analysis showed a slight increase in H3K36me3 levels within gene bodies in H3.3 G34R cells, potentially indicating that the mutation inhibits a H3K36me3 demethylase, not a methyltransferase. Using existing ChIP-seq data for the H3K36me3me2/H3K9me3me2 specific demethylases Kdm4a/b/c they found that H3K36me3 levels are more affected in regions bound by Kdm4a/b/c in H3.3 G34R cells when compared to WT. Similar effects were seen when measuring the H3K9me3 levels in both conditions. The authors showed by peptide pull down that H3.3 G34R binds stronger to Kdm4a/b/c than normal H3 and propose that this affect the activity of Kdm4a/b/c in a dominant negative fashion. They performed RNA-seq of WT and H3.3 G34R cells and overlapped transcriptional changes with the ones observed in Kdm4a/b/c triple KO cells. Here they found that similar genes are affected in H3.3 G34R cells and Kdm4a/b/c triple KO cells, in support of a model in which H3.3 G34R inhibits Kdm4 activity.

The elucidation of the mechanism by which H3.3 G34R can alter gene expression and contribute to development of gliomas is of general interest. Although the results have all been obtained in embryonic stem cells, the manuscript provides some interesting observations, which may have relevance to how H3.3 G34R could contribute to gliomas. One experiment that is missing is to assess whether H3.3 G34R inhibits the catalytic activity of the Kdm4 family in vitro. The authors should provide quantitative experiments to test this. Without this type of experiment, we are unable to recommend publication of the manuscript. Moreover, the authors should address the issues listed below:

1. In general, H3K36me3 and H3K9me3 ChIP-seq should be spiked with *Drosophila* chromatin to get a correct normalization of signals.
2. In Fig 1B, a H3K36me3 and H3K9me3 Western blot should be shown with proper controls, to test if there are global changes in H3K36me3 and H3K9me3 levels.
3. In Fig 1G, Kdm4a, b, c ChIP at the same regions should be performed. What happens to Kdm4? Does it not bind chromatin anymore?
4. In the text, in Fig 1H, I, J and Fig 2D, E, F : The authors should reference in the text and in the figure legend how the Kdm4 or the H3K9me3 /H3K36me ChIP-seq results were obtained.
5. Fig 2D: Kdm4a/c mainly binds at H3K4me3 positive TSS regions. Please make the heat maps centred around H3K4me3 positive regions, as this is where the biggest effect on H3K9me3 levels are observed in Kdm4a/b/c KO cells.
6. Fig 3B and C: A titration experiment should be performed. Here the binding of WT peptide should be done in the presence of increasing amounts of G34R peptide. The authors should also quantify the binding affinities of Kdm4a and Kdm4c with the peptides.
7. Fig 3E: The authors should make an effort to generate quantitative result for the interaction data.
8. Fig 4: The expression data should be quantitatively validated with qRT-PCR for 3-4 selected genes in G34R cells compared to WT cells.

9. The authors should throughout the manuscript write how many times experiments were performed and whether statistics was performed on technical or biological replicates.

Reviewer #2 (Remarks to the Author):

In this manuscript, Voon et al. report that they established a novel knock-in cell line which carries a single copy of H3.3G34R at the endogenous H3F3A locus and identified that H3.3G34R alters the levels of H3K36me3 and H3K9me3 through suppression of KDM4 lysine demethylase family proteins. Biochemical assays and ChIP-seq analyses of H3.3G34R cells and KDM4 triple knockout cells support the interaction of H3.3G34R and KDM4 proteins. Furthermore, RNA-seq data show that the suppression of KDM4 activity by H3.3G34R triggers some transcriptional changes. These findings are previously undescribed and may be relevant to the oncogenic activity of H3.3G34R. However, there are important technical and perhaps conceptual issues in this manuscript that must be addressed in order to lend stronger support to the conclusions.

Major comments

1. The authors claim that the engineered ES-cell line used in the study has single-copy of H3.3G34R; however the information provided is not sufficient. Southern blot data should be shown for the confirmation of heterozygosity as well as the excision of the loxP cassette by Cre recombinase. Southern blot data is also helpful to exclude the possibility of untargeted recombination that could lead to dysregulated expression of H3.3G34R. Furthermore, the chromatogram in Figure 1A is confusing. If the mutation is heterozygous, peaks should be mix of wild-type and mutant alleles. The authors should clearly address these issues.
2. One of the major points of this manuscript is that H3.3G34R has higher affinity with KDM4 proteins that could disrupt the native binding profile. Is the genomic localization of KDM4 proteins altered by H3.3G34R? It is expected that KDM4 proteins are enriched at sites with higher H3.3 occupancy under G34R setting.
3. H3K36me3 mark often shows uneven distribution along the gene body (i.e. lower in TSS side and higher in TTS side). Is the orientation of genes taken into account for the analysis of ChIP-seq data (e.g. Fig. 1C, E, F and Fig. 2D, E)?
4. Does the ectopic expression of H3.3G34R or G34V phenocopy the knock-in model? This could further support the claim that H3.3G34R/V has a dominant negative effect. Currently this claim is not very strongly supported in the manuscript.
5. In Fig. 3B and C, the intensity of flow-through (FT) bands looks almost the same between WT and G34R, though it is supposed to decrease in the G34R setting. In addition, in Fig. 3D, height of HA bands in the lower right blot is different. These points should be explained and/or better image should be provided.
6. In Fig. 3E, western blots for KDM4 proteins should be provided to make sure that the equal amount of KDM4 proteins is precipitated. Also, some bands look saturated. Quantification of band intensity and providing statistics will further strengthen authors' claim.
7. Based on RNA-seq data, genes with lower KDM4 occupancy are most sensitive to H3.3G34R, which does not fit well to the ChIP-seq data that show unchanged H3K36me3 and H3K9me3 level at these sites. The authors should explain this discrepancy
8. An analysis of the transcriptomal profile of H3G34R should be provided. The authors should also include tables of differentially expressed genes or easy access to those.

Minor comments

9. Are WT cells unmodified ES cells, or engineered ES cells with WT H3.3? Does the remaining LoxP site affect the expression of H3.3? In general more detail should provided for the method of generation of the ES cells in use in the manuscript.
10. G34R increased the level of H3K36me3 (Fig. 1G), but decrease that of H3K9me3 (Fig. 2C) at *Srrm2* locus. Is there any explanation for this?
11. Molecular weights should be labeled in western blot images.
12. In Fig. 1H, I and J, what are the gray-tone masks showing?

We would like to thank the reviewers for their helpful comments which have helped to improve this revised manuscript.

Reviewer #1

One experiment that is missing is to assess whether H3.3 G34R inhibits the catalytic activity of the Kdm4 family *in vitro*. The authors should provide quantitative experiments to test this.

Reviewer 1 identified the lack of catalytic data as the single major issue with the previous manuscript. To address this, we performed *in vitro* demethylase assays using KDM4 A/B/C and analysed profiles by mass-spectrometry (Figure 4C). Our results showed that the G34R substitution strongly inhibited the demethylase activity of all three KDM4 A/B/C. This finding lends strong support to our assertion that H3.3 G34R severely impacts the activity of the KDM4 family of demethylases.

In addition to this, the reviewer has asked for other additional technical clarifications which we have attempted to address.

1. In general, H3K36me3 and H3K9me3 ChIP-seq should be spiked with *Drosophila* chromatin to get a correct normalization of signals.

While we agree that a spike-in control would provide some additional normalisation factor, we believe it would be of limited use for the type of chromatin changes which we are trying to detect. The general consensus is that spike-in controls are most useful for studies which anticipate changes in the total signal across the whole genome. Our data and data from other studies (both H3.3 G34R (1) and Kdm4-tKO mutants (2)) indicate that there are no gross alterations in H3K9me3 or H3K36me3 on a genomic scale. We have also included a western blot (Supplementary Figure 2A) which shows that the genomic signal of H3K9me3 and H3K36me3 are not significantly altered in the H3.3 G34R mutant.

It is well-accepted that an input control is sufficient for detecting punctate, localised differences in ChIP-seq signal, such as the changes described in this report. Furthermore, we have also validated these differences using ChIP-qPCR (Figure 2D, 3F) to show that this is a reproducible effect.

2. In Fig 1B, a H3K36me3 and H3K9me3 Western blot should be shown with proper controls, to test if there are global changes in H3K36me3 and H3K9me3 levels.

We have provided this as a supplementary figure (Supplementary Figure 2A). No global alterations in H3K36me3 or H3K9me3 were detected in the G34R mutant which is consistent with observations from other groups (1, 3).

3. In Fig 1G, Kdm4 a, b, c ChIP at the same regions should be performed. What happens to Kdm4? Does it not bind chromatin anymore?

This is a good point. To address this, we determined binding profile of KDM4 by ChIP/qPCR in Flag/bio-tagged KDM4B expressing WT and G34R. We used PCR primers tiled across a gene which had two peaks of KDM4B enrichment with variable H3.3 (G34R) enrichment. We find that KDM4B is retained at the high-H3.3 (G34R) intragenic site but reduced at the H3.3 depleted promoter, in the

G34R mutant compared to WT cells (Figure 4B). This finding agrees with our data that KDM4 preferentially binds H3.3 G34R (Figure 4A). Given the G34R substitution inhibits KDM4 demethylase activity (Figure 4C), it is possible that H3.3 G34R can alter chromatin modifications both directly (e.g. inhibition of KDM4) and indirectly (e.g. redistribution of KDM4). As the regions in Figure 2D are enriched for H3.3 (G34R), it is most likely that the chromatin changes at these sites are due to direct inhibition of KDM4, and the KDM4 profile at these sites is not altered.

4. In the text, in Fig 1H, I, J and Fig 2D, E, F : The authors should reference in the text and in the figure legend how the Kdm4 or the H3K9me3/H3K36me3 ChIP-seq results were obtained.

We apologise for this oversight. The datasets are now clearly referenced in the text and figure legends. Full details can be found in Materials and Methods.

5. Fig 2D: Kdm4a/c mainly binds at H3K4me3 positive TSS regions. Please make the heat maps centred around H3K4me3 positive regions, as this is where the biggest effect on H3K9me3 levels are observed in Kdm4a/b/c KO cells.

While we understand the reviewers' concern, we would prefer to keep the figure centred on H3K9me3 in the manuscript for clarity and consistency. The multi-panel figure contains similar data for H3K9me3 and also H3K36me3. As H3K36me3 is associated specifically with an elongating polymerase, this modification only appears within the gene body and not at H3K4me3 positive promoters. We are concerned that shifting from H3K4me3 enriched promoters for the detection of H3K9me3, to only H3K36me3 enriched sites for detection of H3K36me3, would be unnecessarily confusing for readers.

We have created an alternative figure for the reviewers' perusal, centred on H3K4me3 promoters which gain H3K9me3 in Kdm4-tKO cells. This figure shows that H3K4me3-positive promoters which gain H3K9me3 in Kdm4-tKO cells, also gain H3K9me3 in G34R mutants. We hope this will reassure the reviewer that there are no meaningful differences between centring on H3K4me3 or centring on H3K9me3, but we prefer the latter for simplicity and readability.

6. Fig 3B and C: A titration experiment should be performed. Here the binding of WT peptide should be done in the presence of increasing amounts of G34R peptide. The authors should also quantify the binding affinities of Kdm4a and Kdm4c with the peptides.

We attempted *in vitro* competition assays with synthetic peptides as suggested by the reviewer. However, the WT and H3.3 G34R peptides failed to compete in either direction. This indicates that

the binding of the synthetic H3.3 peptides to KDM4 may be irreversible. We agree with the reviewer that a positive result would have provided unequivocal evidence of a direct preference for H3.3 G34R. However, we do not think that a negative result for the *in vitro* competition assay, necessarily contradicts our *in vivo* observations.

Our *in vivo* binding assays of full-length proteins in a cellular context show preferential interactions between KDM4 and H3.3 G34R (Figure 4A). In addition, the chromatin alterations in both H3K9me3 and H3K36me3 (Figures 2 and 3), as well as our direct evidence that G34R inhibits KDM4 (Figure 4C), strongly suggest that H3.3 G34R influences the activity of KDM4. Nonetheless, we do not claim that this is a comprehensive study on KDM4 and H3.3 interaction and many details surrounding this interaction (in particular, the binding of KDM4 to H3.3 and release from the binding) still need to be determined as part of a further follow-up study.

7. Fig 3E: The authors should make an effort to generate quantitative result for the interaction data.

We agree with the reviewer and apologise for the oversight. This has been supplied as bar charts in Fig. 4A and as a supplementary figure (Fig. S4A) of three independent experiments.

8. Fig 4: The expression data should be quantitatively validated with qRT-PCR for 3-4 selected genes in G34R cells compared to WT cells.

We agree with the reviewer and apologise for the oversight. We have now validated expression changes of a number of up- and down-regulated genes by qRT-PCR and the results are shown in Supplementary Figure 3D.

9. The authors should throughout the manuscript write how many times experiments were performed and whether statistics was performed on technical or biological replicates.

We agree with the reviewer and apologise for the oversight. This is now clearly indicated in the figure legends and in the materials and methods. However, we have opted for the conventional term of “independent experiments” as we repeated entire experiments in cell lines which are not necessarily considered “biological” replicates.

Reviewer #2

1. The authors claim that the engineered ES-cell line used in the study has single-copy of H3.3G34R; however the information provided is not sufficient. Southern blot data should be shown for the confirmation of heterozygosity as well as the excision of the loxP cassette by Cre recombinase. Southern blot data is also helpful to exclude the possibility of untargeted recombination that could lead to dysregulated expression of H3.3G34R. Furthermore, the chromatogram in Figure 1A is confusing. If the mutation is heterozygous, peaks should be mix of wild-type and mutant alleles. The authors should clearly address these issues.

We apologise for not providing sufficient information or clarity in the original manuscript. We have substantially revised the manuscript to clearly address these issues. The revised figure shows confirmation of targeting, heterozygosity, and excision, by Southern blot as requested (Figure 1B).

The chromatogram does not show heterozygosity as the sequencing primers were specific to the H3.3 G34R mutant transgene, and do not detect WT H3.3. We have clarified this in the manuscript.

We have also included a schematic to clearly depict that there are two genes (four copies) which encode H3.3 (H3f3a and H3f3b) in the mouse and human genomes. Our cell model targets a single copy of H3f3a, which recapitulates the mutations found in paediatric gliomas (4). The remaining single WT copy of H3f3a and two WT copies of H3f3b continue to be expressed. We have included a schematic of RNA-seq reads showing expression of H3.3 G34R alongside WT H3f3a and H3f3b (Fig. 1D in the revised manuscript). The full reads are also included as supplementary data (Supplementary Figure 1 in the revised manuscript).

We hope these revisions will help clarify the cell model and clearly show expression of a single copy of H3f3a G34R, alongside WT H3f3a and WT H3f3b.

2. One of the major points of this manuscript is that H3.3G34R has higher affinity with KDM4 proteins that could disrupt the native binding profile. Is the genomic localization of KDM4 proteins altered by H3.3G34R? It is expected that KDM4 proteins are enriched at sites with higher H3.3 occupancy under G34R setting.

This is a good question which we have addressed by CHIP of Flag/bio-tagged KDM4B. To test for H3.3-dependent localisation of KDM4, we used PCR primers tiled across a gene which had two peaks of KDM4B enrichment with variable H3.3 (G34R) enrichment (Figure 4B). The reviewer is correct that H3.3 G34R changes the localisation of KDM4, and KDM4 enrichment in the G34R mutant is dependent on the presence of H3.3.

3. H3K36me3 mark often shows uneven distribution along the gene body (i.e. lower in TSS side and higher in TTS side). Is the orientation of genes taken into account for the analysis of ChIP-seq data (e.g. Fig. 1C, E, F and Fig. 2D, E)?

The reviewer is correct that there is bias in H3K36me3 across the length of a gene, however, our analyses were centred on H3K36me3 peaks, rather than across the gene length. This was necessary as many of the alterations occurred as punctate, localised gains in H3K36me3 (e.g. Figure 2G, 3E). If we scanned across a gene, we would have missed many of these most prominent localised changes. The uneven distribution of H3K36me3 does not affect our results as these are centred on H3K36me3,

and assuming there are an equal number of positive and negative oriented genes, there would be equal bias in both directions.

4. Does the ectopic expression of H3.3G34R or G34V phenocopy the knock-in model? This could further support the claim that H3.3G34R/V has a dominant negative effect. Currently this claim is not very strongly supported in the manuscript.

We have chosen to use a knock-in model as our single-copy H3f3a G34R model closely reflects the mutation found in paediatric gliomas and we are concerned that an ectopic expression of H3.3 G34R would not be informative with respect to human cancers.

To provide evidence that our H3.3 G34R knock-in model expresses WT H3.3 robustly, we have now included RNA-seq data showing that WT H3f3b, WT H3f3a and H3f3a G34R transcripts are all expressed (Figure 1D). The RNA-seq reads also provide an approximation of the relative proportion of each transcript, and is consistent with the presence of a single copy of H3.3 G34R. Therefore, the phenotype we observe are almost certainly due to a dominant-negative effect. We hope this helps address the reviewers' concerns.

5. In Fig. 3B and C, the intensity of flow-through (FT) bands looks almost the same between WT and G34R, though it is supposed to decrease in the G34R setting. In addition, height of HA bands in the lower right blot is different (Fig. 3D in the original manuscript). These points should be explained and/or better image should be provided.

In vitro binding experiments require that the binding partner (e.g. KDM4 protein) be present in excess, and only a very small proportion of the total protein is bound. This ensures that the binding partner (KDM4) is not a limiting factor in the binding assays. In general, only a small fraction of the excessive amount of flow-through is loaded for gel electrophoresis and western blot analysis. Thereby, we would not expect the flow-through to decrease in the G34R setting.

Many factors can influence the way which a protein gel runs (e.g. imperfections in the gel, particulates in the running buffer or wells, uneven electrodes etc). To assure the reviewer that the small difference in protein migration in the two wells are not meaningful, we have provided the complete gel with the relevant lanes outlined to red (Supplementary Figure 4B in the revised manuscript).

6. In Fig. 3E, western blots for KDM4 proteins should be provided to make sure that the equal amount of KDM4 proteins is precipitated. Also, some bands look saturated. Quantification of band intensity and providing statistics will further strengthen authors' claim.

We have provided anti-flag western blots for KDM4 (Supplementary Figure 4 C-E) which validate our findings that KDM4 binds H3.3 G34R with higher affinity compared to WT H3.3. We have also quantified band intensity (Figure 4A) and provided statistics across three independent experiments (Supplementary Figure 4A).

7. Based on RNA-seq data, genes with lower KDM4 occupancy are most sensitive to H3.3G34R, which does not fit well to the CHIP-seq data that show unchanged H3K36me3 and H3K9me3 level at these sites. The authors should explain this discrepancy.

An analysis of the transcriptomal profile of H3G34R should be provided. The authors should also include tables of differentially expressed genes or easy access to those.

As we have extensively revised and improved the manuscript according to reviewer suggestions, we no longer think that the original transcriptional data fit well with the revised manuscript. We have now provided strong data which details molecular interactions between H3.3 G34R and KDM4, and we would like to focus on these aspects.

We have not carried out the extensive phenotypic analyses which would be required to dissect the exact ways in which KDM4 (and H3.3 G34R) controls gene expression. We think this is a topic which deserves full and detailed investigations, and would be beyond the scope of this study.

We have retained the transcriptional analyses which illustrate the similarities between the H3.3 G34R mutant and KDM-tKO mutants (Figure 3G-L in the revised manuscript) as these findings reinforce the core message of this study. We have also included a table of differentially expressed genes as requested (Supplementary Table 3). In addition, all data and analyses will be made publically accessible for any follow-up studies.

Minor comments

8. Are WT cells unmodified ES cells, or engineered ES cells with WT H3.3? Does the remaining LoxP site affect the expression of H3.3? in general more detail should provided for the method of generation of the ES cells in use in the manuscript.

We apologise for this oversight. The WT cells are unmodified ES cells. This has now been explicitly stated in the methods.

We cannot be absolutely certain that the LoxP site does not affect expression of H3.3. However, our RNA-seq data suggests that any abnormalities in expression are exceedingly mild and small decreases in H3.3 have virtually no detectable phenotypic effects (5).

We have also provided more detailed methods as requested.

9. G34R increased the level of H3K36me3 (Fig. 1G), but decrease that of H3K9me3 (Fig. 2C) at Srrm2 locus. Is there any explanation for this?

It may be possible that KDM4 can retain some activity against H3K9me3 under some circumstances. However, this is purely speculative and requires much deeper investigations which are beyond the scope of this study. We considered it important to present the data as completely and honestly as possible even if we cannot yet explain all our observations.

10. Molecular weights should be labeled in western blot images.

We apologise for the oversight. This has been corrected.

11. In Fig. 1H, I and J, what are the gray-tone masks showing?

The figures have been updated to remove the gray-tone masks. They were originally intended to mask KDM4 promoters and draw the readers' attention to the gene body peaks of KDM4 and H3K36me3.

1. P. W. Lewis, M. M. Muller, M. S. Koletsky, F. Cordero, S. Lin *et al.*, Inhibition of PRC2 activity by a gain-of-function H3 mutation found in pediatric glioblastoma. *Science* **340**, 857-861 (2013).
2. M. T. Pedersen, S. M. Kooistra, A. Radziszewska, A. Laugesen, J. V. Johansen *et al.*, Continual removal of H3K9 promoter methylation by Jmjd2 demethylases is vital for ESC self-renewal and early development. *The EMBO journal* **35**, 1550-1564 (2016).
3. K. M. Chan, D. Fang, H. Gan, R. Hashizume, C. Yu *et al.*, The histone H3.3K27M mutation in pediatric glioma reprograms H3K27 methylation and gene expression. *Genes & development* **27**, 985-990 (2013).
4. J. Schwartzenuber, A. Korshunov, X. Y. Liu, D. T. Jones, E. Pfaff *et al.*, Driver mutations in histone H3.3 and chromatin remodelling genes in paediatric glioblastoma. *Nature* **482**, 226-231 (2012).
5. M. C. Tang, S. A. Jacobs, D. M. Mattiske, Y. M. Soh, A. N. Graham *et al.*, Contribution of the two genes encoding histone variant h3.3 to viability and fertility in mice. *PLoS genetics* **11**, e1004964 (2015).

REVIEWERS' COMMENTS:

Reviewer #1 (Remarks to the Author):

In the revised version the authors have addressed all our comments and we can recommend publication of the manuscript. However, we suggest that the authors do not write that H3G34R has high affinity of KDM4 in the abstract. The authors do not measure affinities in the manuscript. They could perhaps write that H3G34R show stronger binding to KDM4 than wild type H3.

Reviewer #2 (Remarks to the Author):

No further comments.

REVIEWERS' COMMENTS:

Reviewer #1 (Remarks to the Author):

In the revised version the authors have addressed all our comments and we can recommend publication of the manuscript. However, we suggest that the authors do not write that H3G34R has high affinity of KDM4 in the abstract. The authors do not measure affinities in the manuscript. They could perhaps write that H3G34R show stronger binding to KDM4 than wild type H3.

We have removed all references to “affinity” and replaced this with the term “preferential binding” in the abstract and throughout the manuscript.

Reviewer #2 (Remarks to the Author):

No further comments